# Retrospective Evaluation of 20 Years of Outpatient Dental Services to Adults with Disabilities at the Dental Hospitals of the Medical University of Innsbruck, Austria

**DOI:** 10.3390/healthcare12050503

**Published:** 2024-02-20

**Authors:** Dagmar Schnabl, Matthias Michael Strohm, Pit Eugene Schummer, Lukas Sigwart, Ines Kapferer-Seebacher

**Affiliations:** 1University Hospital of Dental Prosthetics, Medical University of Innsbruck, 6020 Innsbruck, Austria; m.strohm@kabelmail.de (M.M.S.);; 2University Hospital of Conservative Dentistry and Periodontology, Medical University of Innsbruck, 6020 Innsbruck, Austria; lukas.sigwart@tirol-kliniken.at (L.S.); ines.kapferer@i-med.ac.at (I.K.-S.)

**Keywords:** special needs adults, chairside dental treatment, dental students

## Abstract

Disabled persons’ chairside dentistry is challenging. We aimed for a retrospective breakdown of dental services delivered to disabled patients by dental students and to discuss feasibility of a chairside approach. Consecutive patients, who received scheduled dental treatment by dental students from 2002 to 2021, were included. Demographic data, medical diagnoses, number of treatment sessions, performed treatments, and treatment break-offs were collected and analyzed with descriptive statistics. In total, 224 individuals with various disabilities (mean age 36.4 ± 14.6 years) received dental services in 2282 sessions altogether (10.3 ± 11. sessions per patient). Professional tooth cleaning was the most frequently provided treatment (55.8% of sessions). A total of 654 teeth were restored with fillings, 97 teeth were extracted, 56 teeth had endodontic treatment, and 25 removable dentures were fitted. Treatment break-off due to incompliance and referral to dental general anesthesia occurred in 74 patients (33%). Chairside treatment of disabled persons by dental students is feasible in many cases. Our study may serve as an incentive for clinicians/researchers to report on treatment modalities and outcomes of chairside dentistry in patients with special oral health care needs, preferably by the use of prospective study designs, to contribute data and strategies in the fight for control of oral health inadequacies.

## 1. Introduction

Deficient oral homecare and limited access to dental care have been identified as causes for a backlog in oral health in persons with disabilities [1,2]. Biological conditions, as well as organizational requirements, demand efforts of both caregivers and dental professionals [3,4]. According to the nature of disablement (physical, intellectual, psychological, or combined), appointment agreement, transportation, and access to dental offices require commitment and support [3,4,5,6,7]. In practice facilities, certain construction conditions, such as a wheelchair ramp, an elevator/chair lift, or disabled toilets are requisite.

While dental treatment under general anesthesia (usually performed in specialized centers) is a widely practiced and reported strategy in the management of special needs patients [8,9,10], studies on chairside dental therapy of disabled individuals under office conditions are extremely scarce. Auerbacher et al. (2023) [11] conducted a retrospective feasibility study on chairside oral prophylaxis for people with profound intellectual or multiple disabilities. Authors concluded that an individualized and disability-specific treatment strategy using various non-invasive and non-pharmacological behavioral guiding techniques effected an increase in compliance to professional tooth cleaning and reduced the need for dental general anesthesia. More invasive chairside treatment, such as restorative therapy, was not attempted. A Brazilian feasibility study compared conventional treatment to atraumatic treatment for disabled persons in a chairside setting [12]. Atraumatic treatment by use of use of hand instruments and glass-ionomer restorations proved to be far more feasible than conventional treatment by the use of rotary instruments and composite.

Generally, a well-trained and geared staff constitute the basis for appropriate communication and confidence building, which are essential in the attainment of collaboration (within patients’ personal limits) [13]. Additionally, alternative approaches, e.g., behavior management, hypnodontia, or conscious sedation might increase treatment success [14,15]. Aside from these general professional framework conditions, patience, dedication, and the arrangement of enough time seem to be crucial factors in the achievement of amenability and compliance in patients with disablement [13,16].

At the University Hospital of Innsbruck, in the course of clinical training, dental students have delivered scheduled outpatient dental treatment to patients with intellectual, physical, psychological, or combined impairment for several decades.

The aim of this retrospective cohort study was a breakdown of chairside dental services to disabled adults by dental students over a period of 20 years in order to evaluate the performance-related outcomes of our outpatient program and to discuss feasibility of disabled individuals’ chairside treatment. Our findings contribute clinically relevant data to the still sketchy knowledge on disabled persons’ dental care needs as postulated by Ferreira et al. 2023 in a recent review, which evaluated scientific production on ““disabled persons” and “dentistry”” over a 20 year period [17]. Most included publications were cross-sectional studies and simple literature studies, results of which are not directly applicable in clinical practice. Authors suggested that future research trends should focus on behavior guidance, dental education, and access to dental services. Our results might serve as an incentive for clinicians and researchers to report on conditions, treatment modalities, and outcomes of chairside dentistry in patients with special oral health care needs.

## 2. Materials and Methods

### 2.1. Ethical Approval and Registration

This study was carried out in accordance with the 1964 Declaration of Helsinki and its later amendments. Ethical approval was obtained prospectively by the Ethics Committee of the Medical University of Innsbruck, study number 1360/2021. This study was registered at the registry for clinical studies of the University Hospital of Innsbruck (Koordinationszentrum für klinische Studien; kks-innsbruck@i-med.ac.at), registration number 20220124-2798.

### 2.2. Subjects

All (consecutive) patients who received scheduled dental treatment in the setting of the outpatient students’ program for disabled persons at the dental hospitals of Innsbruck from 1 January 2002 to 31 December 2021 were included.

Inclusion criteria for participation in the program were intellectual, physical, psychological, or combined disablement of Cooperation Level Scale grades 4 to 6 [1] that were judged not so severe as to necessitate immediate referral to dental general anesthesia, but seemed worth attempting a chairside approach and legal guardians’ written informed consent. Exclusion criteria were severe disablement implicating non-cooperation or defense (corresponding to Cooperation Level Scale grades 0 to 2 [1]) and extensive dental treatment demand that was not deemed manageable under office conditions by the respective consultant in charge.

### 2.3. Data Acquisition

Demographic and medical data were retrospectively derived from electronic patient files (ClinicWare NICE, v0436.0040, Agfa Healthcare, Bonn, NRW, Germany) and tabularized in a pseudonymized manner (Excel, Microsoft, Redmont, WA, USA).

The following data were recorded:Date of birthGenderMode of residence and caretakership-Private home and private care-Private home and day-care center-(Public) residential home and careDate of each scheduled or unscheduled sessionProvided dental treatment per session-Professional tooth cleaning (polishing), including oral hygiene (re-) instruction according to requirements-Removal of calculus-Periodontal therapy including deep scaling-Restorative therapy: Fillings per tooth and tooth surfaces (one to five surfaces per tooth) by materials (composite, amalgam, or high-viscosity glass ionomer cement)-Tooth extractions-Endodontic treatment-Provision of removable dentures (partial or complete) and aftercare measures-Provision of fixed prosthetic restorationsTreatment break-off due to non-compliance

### 2.4. Statistical Analysis

Data was analyzed by means of descriptive statistics. Numerical data were reported as mean ± standard deviation (SD) and categorical data were summarized as absolute and relative frequencies. All calculations were performed in SPSS software (SPSS Statistics Version 27.0, IBM, Armonk, NY, USA).

## 3. Results

### 3.1. Subjects

In the surveyed time interval of 20 years, 224 individuals with disabilities received dental services by dental students in altogether 2282 sessions. In total, 102 patients (45.5%) were female and 122 (54.5%) were male. Mean age at first presentation was 36.4 ± 14.6 years.

In 157 recruited patient files (70.1%) one or more explicit medical diagnosis/diagnoses was/were documented, whereas no specific diagnosis (clarifying the nature of disablement) was recorded in 67 files (29.9%). Of the 157 patients with documented diagnoses, 39 (24.8%) were diagnosed with multiple disabilities/medical disorders, resulting in a total of 198 medical diagnoses in 157 patients, please see Table 1.

In 59 individuals (37.6% of individuals with documented diagnoses), “intellectual disablement” was explicitly reported. However, many of the documented disorders, diseases, or syndromes assessed in participants of this study implicate intellectual disablement.

A total of 113 (50.5% of 224 patient files) contained no information on the mode of residence or caretakership. Of the 111 persons with known caretakership, 61 patients (55%) were cared for at a residential home, 48 (43.2%) were cared for at a private home, and 2 patients (0.9%) lived at a private home but received institutional day-care.

### 3.2. Number and Distribution of Treatment Sessions

The mean number of sessions per patient was 10.3 ± 11.2 (range one to 67) within the surveyed period of 20 years. A number of 31 patients (13.8%) made use of one single session and 27 patients (12.1%) received treatment twice. 90 patients (40.3%) had less than five and 31 patients (14%) had 20 or more sessions. The time frame between the first and the last appointment per patient averaged 60.7 ± 63.5 months (range zero to 240 months). In 78 patients (35%) the stretch of treatment time amounted to 12 months or less, in 88 patients (39%) to more than 60 months, and in 46 patients (20.5%) to more than 120 months.

A number of 1894 (83% of 2282) sessions were scheduled Tuesday morning sessions, whereas 165 (7.2%) were unscheduled emergency sessions addressing acute complaints.

In total, 30 patients (13.4%) were ascribed to treatment by a constant student in altogether 223 sessions (9.8%), amounting to 5.9 sessions per patient on average.

### 3.3. Provided Treatment

#### 3.3.1. Oral Hygiene Measures and Periodontal Therapy

In 1273 treatment sessions (55.8%), professional tooth cleaning (including toothbrushing instruction as needed) was performed. In 1048 sessions (45.9%) calculus was removed, and in 69 sessions (3%) periodontal therapy (deep scaling) was provided.

#### 3.3.2. Restorative Therapy

Within the study period of 20 years, 654 teeth were restored with fillings, 358 (54.7%) of which maxillary and 296 (45.3%) mandibular. Figure 1 displays the distribution of restored teeth. 554 teeth (84.7%) were restored in the course of scheduled Tuesday sessions by changing students on duty, 77 (11.8%) in scheduled “non-Tuesday” sessions performed by students constantly assigned to the respective patient, and 23 (3.5%) in non-scheduled emergency sessions. 454 teeth were restored with composite, 109 with amalgam, and 91 with temporary filling materials. The extension of fillings varied from one surface to five surface restorations as depicted in Table 2.

#### 3.3.3. Tooth Extractions

Over 20 years, 97 teeth (56 (57.7%) maxillary and 41 (42.3%) mandibular) were extracted. Figure 2 illustrates the distribution of extracted teeth.

In total, 48 (49.5%) extractions were performed in an emergency setting, 40 (41.2%) in scheduled sessions, and 9 (9.3%) in “non-Tuesday” sessions.

#### 3.3.4. Endodontic Treatment

Altogether, 128 (5.6% of 2282) sessions were used for root canal treatment in altogether 56 teeth. In total,79 sessions were used to perform endodontic treatment of 34 maxillary and 49 sessions were used for endodontic treatment of 22 mandibular teeth. Distribution of endodontically treated teeth and number of treatment sessions per tooth number, as shown in Table 3. Out of all sessions, 79 (57.8%) were scheduled treatment sessions, 36 (28.1%) were sessions carried out by students who were assigned for endodontic treatment of the respective tooth, and 18 (14.1%) sessions were unscheduled pain treatment sessions. On average, per tooth, 2.3 sessions were spent for endodontic therapy.

#### 3.3.5. Prosthodontics

Over the study period, five complete maxillary dentures were fitted in “non-Tuesday” sessions by students who were allocated to prosthodontic treatment of the respective patient. Thirteen metal framework partial dentures (seven maxillary and six mandibular, twelve of which carried out in “non-Tuesday” sessions) and five acrylic mucosa-borne partial clasped dentures (four maxillary and one mandibular) were delivered. The mean numbers of required sessions per type of removable denture are displayed in Table 4. Five permanent denture linings, and 18 denture repairs were provided. Denture relieving was documented 17 times. No fixed restorations were made.

### 3.4. Treatment Break-Off

In total, 74 patients (33%) of the outpatient students’ program for disabled persons were referred to dental general anesthesia due to insufficient compliance/cooperation. Thereby, outpatient treatment was either stopped or was resumed after rehabilitation under general anesthesia in the sense of maintenance and preventive care. A number of 51 patients underwent dental general anesthesia once; 13 patients twice; 4 patients 3 times; 3 patients 4 times; 2 patients 5 times; and 1 patient 6 times. In cases of multiple treatment under general anesthesia, intervals between treatment sessions ranged from one to eleven years.

## 4. Discussion

In special care dentistry, the provision of routine dental care is complicated by limitations relating to communication, cooperation, health conditions, and social context [18,19]. Therefore, in private practices, chairside treatment of patients with special needs is a rather challenging procedure that demands high expertise, manpower costs and time investment, which are usually not adequately resourced by public healthcare insurances. In hospital settings, dental treatment of persons with disablement is frequently carried out under general anesthesia within a (day-unit) hospital stay rather than in terms of an outpatient approach on the dentist’s chair [8,9,10]. Dental general anesthesia seems well justified in patients displaying extensive dental treatment demand combined with severe disablement, incomprehension, and defense (Cooperation Level Scale grades 0 to 3 [1]). However, in patients with minor dental complaints and less severe disablement (Cooperation Level Scale grades 4 to 6 [1]) a chairside treatment approach seems worth attempting. At the University Hospital of Innsbruck, in the course of supervised clinical training, dental students enrolled in the clinical phase of education (8th to 12th semester) have delivered scheduled outpatient dental services to patients with intellectual, physical, psychological, or combined impairment for many years. Each Tuesday, two patients with various disabilities, who have presented on private or public caretakers’ initiative or have been referred to the hospital by general practitioners, medical specialists, or dentists in practice, are scheduled. At admission, demographic data and the medical history (including medical diagnoses and medication) are recorded. Whenever possible, an orthopantomogram is taken. By means of an initial clinical assessment with regard to the severity of disablement (cooperation level) and the nature and extent of dental treatment demand, individuals with disablement are assigned to the hospital’s outpatient students’ program if chairside treatment is deemed feasible, or referred to dental general anesthesia, if patients are judged incapable of cooperation with standard dental treatment [20,21]. At the second appointment, according to clinical and radiological findings, a treatment plan is elaborated, usually consisting of (initial and repeated) oral hygiene instructions as needed, professional tooth cleaning, and, if necessary, periodontal, restorative, or endodontic treatment, tooth extractions, and (mostly removable) prosthetics. This treatment plan is carried out in subsequent sessions by changing students on duty. After its completion, further appointments are scheduled in longer intervals in terms of supportive care.

In some exceptions, for the provision of more complex treatment procedures such as root canal treatment or denture fitting, patients are temporarily assigned to one constant student and receive extra appointments.

In case of dental emergencies, e.g., pain or loose or lost restorations, (unscheduled) emergency treatment is available to clients with disablement, which is carried out by dental students or residents in charge.

Altogether, 244 patients were managed over the study period of 20 years. Patients were fostered by either private or professional caretakers, who arranged for appointments and transportation to and from outpatient treatment. Age distribution (mean 36.4 ±14.6 years) and the outline of diagnoses illustrate the presence of rather congenital or developmental disabilities than that of age-related disorders in the study population. In total, 1894 scheduled Tuesday sessions per 20 years correspond to barely two treatment sessions per workweek or treatment day (9:00 o’clock until noon), allowing an ample time span for each patient. On average, patients received 10.2 treatment sessions within a time frame of 60.7 ± 63.5 months.

In the investigated study cohort, intellectual disablement (frequently in combination with physical impairment) was the most prevalent diagnosis. A limited ability to cooperate with oral care/dental treatment, hampered access to dental care, and a complaint-driven rather than preventive utilization of dental services may be reasons for poorer oral hygiene, poorer periodontal status conditions, and higher unmet caries treatment needs in individuals with intellectual disabilities [1,22].

Preventive measures such as professional toothcleaning with or without toothbrushing instructions (performed in 1273 (55.5%) treatment sessions), and scaling (carried out in 1048 (45.9%) sessions) represented the most frequent treatments applied in our study. This reflects the general necessity for intense prophylaxis and improvement of oral care in patients with disabilities, as asserted in a recent review by Molina et al. (2022) [23]. A retrospective German feasibility study showed that the implementation of various behavioral guidance techniques and communication strategies in a supportive environment enabled even patients with severe disabilities to receive chairside professional toothcleaning [11].

Periodontal therapy (deep scaling) was provided in only 69 sessions (3%), probably due to young age of the study group with periodontitis not (yet) present, reduced cooperation of disabled patients with tedious therapy, or the rather generous indication of tooth extraction in case of (severe) periodontitis in the presence of limited oral homecare.

Scientific publications on more invasive chairside dental treatments in disabled individuals are extremely rare. Restorative and endodontic therapy or tooth extractions are frequently carried out under general anesthesia. Molina et al. (2015) conducted a feasibility study that compared conventional restorative treatment (use of rotary instruments, composite, and rubber dam) to atraumatic restorative treatment (use of hand instruments and high-viscosity glass-ionomer) in disabled persons in a chairside setting [12]. Authors showed that atraumatic treatment was highly accepted and feasible in 79% of patients (*n* = 47), while the conventional treatment option was less feasible (33%; *n* = 7). A five-year follow-up study evaluated restoration survival and confirmed that atraumatic restorative treatment using glass ionomer was equally effective as conventional treatment with composite [24]. However, to the authors of this study, some doubt remains as to whether carious lesions forming extensive undercuts are sufficiently excavatable with hand instruments only or whether severely destructed teeth are adequately restorable with glass-ionomer.

In our clientele, altogether 654 teeth were restored. Restorative therapy was mostly (in 69.4% of filled teeth) accomplished with composite whenever dry conditions were contrivable. Unfortunately the use of rubber dam was not (consistently) documented in patient files. The disposition of amalgam was limited to cavities with subgingival extension in vital posterior teeth where only relatively dry conditions where achievable. 13.9% of teeth were restored with glass ionomer cement. The reason for the use of a temporary material instead of permanent materials might have been temporary sealing during endodontic treatment, insufficient cooperation, or the unfeasibility of provision of permanent restorations, or a lack of time. Extensive (four and five surface) restorations were accomplished in 8.6% of teeth. Although both prepared cavities and completed fillings were supervised by the respective consultant in charge, the quality of restorations in terms of border seal, anatomical design and elaboration, approximal and occlusal contact, or durability was not evaluable within the retrospective study design.

During the study period of 20 years, almost 100 teeth were extracted, about half of which in terms of unscheduled emergency treatment (for pain). The distribution of extracted teeth seems erratic (Figure 2), just as the distribution of restored teeth (Figure 1). Teeth 22 and 36 were the most frequently (six times) and teeth 43 and 47 the least frequently (zero times) extracted teeth. Whether caries and its sequelae or periodontal disease represented the indication for tooth extraction was not (systematically) available from patient documentations. Generally, extractions of compromised teeth are probably carried out more expeditiously in persons with disablement than in non-disabled individuals in order to effect rapid and durable pain relief and to avoid straining treatment sessions. However, extensive extraction treatment in preference to restorative dentistry seems not advisable and does not preempt further need for dental therapy [25].

Endodontic treatment was performed in 56 teeth in 128 sessions. Alas, success of root canal treatment with respect to (long-term) pain relief or tooth survival time was not assessable.

In total, 23 out of 224 individuals aged 36.4 ± 14.6 years were provided with removable total or partial dentures. Thereby, most complete or metal framework prostheses were fitted by students permanently assigned to patients for the duration of denture fitting. Confidence building and continuous management by one and the same person/student seems to be a favorable strategy in the performance of complex dental procedures. The numbers of sessions (impression taking to delivery) needed for complete or metal framework dentures (please see Table 4) appear to correspond to those usually spent in denture fitting in non-disabled persons.

This study has shown that chairside treatment of individuals with disablement is contrivable in many cases. Even dental students, who have little experience and currently receive no explicit training in the management of special needs patients, are able to cope with the task on condition of supervision and the accordance of enough time. However, insufficient compliance and defense (frequently resulting from problems coping with noise and vibration of the drill [23]) are limiting factors that may lead to treatment-break-off and assignment to dental general anesthesia as was the case in one third of the study population. Of these 74 (33%) individuals, two thirds underwent multiple treatment sessions under general anesthesia. The resumption of chairside prophylaxis in terms of follow-up sessions after rehabilitation under general anesthesia seems essential to maintain oral health or to recognize further treatment demand and initiate therapy [26,27].

## 5. Limitations

Limitations of the present study are owed to its retrospective design, which merely allowed quantitative assessment of provided treatment but could not render data on treatment outcomes in terms of quality or short- and long-term success or failure.

## 6. Conclusions

Sparce data is available on the feasibility of chairside dental management of disabled persons. Our study showed that chairside treatment of patients with disablement (Cooperation Level Scale grades 4 to 6 [1]) by dental students is feasible and beneficial in many cases under the premises of supervision by consultants, individualized treatment planning, and the arrangement of lavish time. The most common medical diagnosis in the study collective (54.5% males and 45.5% females; mean age 36.4 ± 14.6 years) was intellectual disablement, alone or as part of complex disorders or syndromes. Professional toothcleaning/scaling was the most frequently delivered treatment. Also, restorative therapy, endodontic treatment, tooth extractions, and denture fitting were provided. The provision of multistep dental or prosthodontic procedures by the very same person proved to be favorable with respect to confidence building. In case of treatment break-off because of incompliance, dental general anesthesia remained a backup plan. After rehabilitation under general anesthesia or by chairside treatment, recall sessions and prophylaxis were resumed to maintain oral health or to recognize further treatment demand.

Disabled persons’ oral health inadequacies must be addressed. Our study shall serve as an incentive for clinicians and researchers to report on conditions, treatment modalities, and outcomes of chairside dentistry in patients with special oral health care needs, preferably by the use of prospective study designs. Dental school directors and teachers in particular are encouraged to present their teaching concepts and organizational and treatment strategies in the management of disabled patients. Practice-oriented treatment guidelines might be developed. Dental training should focus more on special care dentistry.

## Figures and Tables

**Figure 1 healthcare-12-00503-f001:**
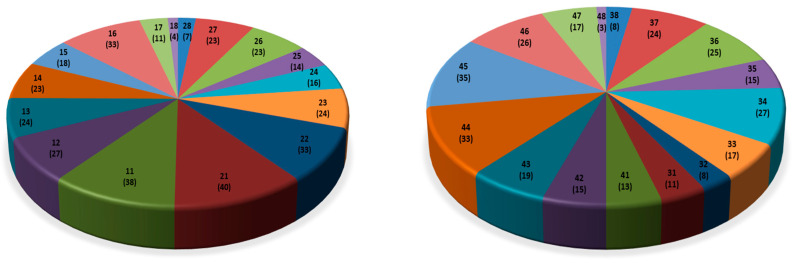
Distribution of restored maxillary (18 to 28; *n* = 358) and mandibular teeth (38 to 48; *n* = 296).

**Figure 2 healthcare-12-00503-f002:**
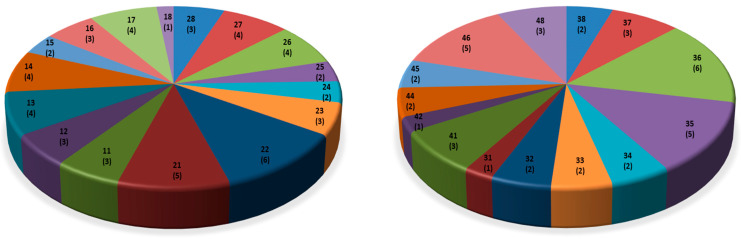
Distribution of extracted maxillary (18 to 28; *n* = 56) and mandibular teeth (38 to 48; *n* = 41).

**Table 1 healthcare-12-00503-t001:** Medical Diagnoses in patients who received dental care in the course of the students’ outpatient treatment program for adults with disablement. Due to multimorbidity in 39 patients (24.8%), 198 medical diagnoses were ascribed to 157 patients.

Diagnosis	Number (%)
Intellectual disablement (not further specified)	59 (29.8)
Epilepsy/seizure disorders	36 (18.2)
Down’s syndrome	14 (7.1)
Status post cerebral apoplexia/hemorrage/ischaemia	14 (7.1)
Spastic disorders	13 (6.5)
Physical disablement (not further specified)	12 (6)
Cerebral maldevelopment/developmental delay	10 (5)
Autism	9 (4.5)
Speech impediment	4 (2)
Ataxia	3 (1.6)
Dementia	3 (1.6)
Schizophrenia	3 (1.6)
Dysphagia	2 (1)
Psychological impairment (not further specified)	2 (1)
Neuropathy	2 (1)
Multiple sclerosis	2 (1)
Visual impairment	2 (1)
Autoaggression	2 (1)
Apallic syndrome	2 (1)
Anxiety disorder	1 (0.5)
CCFDN syndrome *	1 (0.5)
Chorea Huntington	1 (0.5)
Parkinson’s disease	1 (0.5)
Total	198 (100)

* CCFDN, congenital cataract facial dysmorphism neuropathy syndrome.

**Table 2 healthcare-12-00503-t002:** Breakdown of fillings by materials and extension (one to five tooth surfaces).

Filling Material	Number (Percentage) of Restored Teeth	Number (Percentage) of One-Surface Fillings	Number (Percentage) of Two-Surface Fillings	Number (Percentage) of Three-Surface Fillings	Number (Percentage) of Four-Surface Fillings	Number (Percentage) of Five-Surface Fillings
Composite	454 (69.4%)	281 (43%)	66 (10.1%)	60 (9.2%)	8 (1.2%)	39 (5.9%)
Amalgam	109 (16.7%)	50 (7.6%)	23 (3.5%)	27 (4.1%)	1 (0.2%)	8 1.3%)
Glass- ionomer	91 (13.9%)	56 (8.6%)	24 (3.7%)	11 (1.6%)		
Total	654 (100%)	387 (59.2%)	113 (17.3%)	98 (14.9%)	9 (1.4%)	47 (7.2%)

**Table 3 healthcare-12-00503-t003:** Distribution of endodontically treated mandibular (18–28) and maxillary teeth (38–48).

Tooth Number	18	17	16	15	14	13	12	11	21	22	23	24	25	26	27	28
Number of treatment session	0	3	10	1	3	11	4	12	6	5	1	11	7	0	5	0
Number of endodontically treated teeth	0	2	3	1	1	3	3	6	4	3	1	4	2	0	1	0
Number of endodontically treated teeth	0	0	3	1	3	1	0	1	3	2	2	1	2	3	0	0
Number of treatment session	0	0	5	1	6	2	0	3	17	4	2	1	2	8	0	0
**Tooth number**	**48**	**47**	**46**	**45**	**44**	**43**	**42**	**41**	**31**	**32**	**33**	**34**	**35**	**36**	**37**	**38**

**Table 4 healthcare-12-00503-t004:** Register of removable dentures and mean ± standard deviation number of required treatment sessions.

Removable Denture	Number	Number of Treatment Sessions (Mean ± Standard Deviation)
Full upper denture	5	6.60 ± 1.52
Metal framework upper partial denture	4	3.50 ± 1.29
Metal framework lower partial denture	1	6
Acrylic upper partial denture	7	5.14 ± 1.35
Acrylic lower partial denture	6	5.50 ± 1.05

## Data Availability

All data generated or analyzed during this study are included in this published article.

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
