# Peer review of "Retrospective Evaluation of 20 Years of Outpatient Dental Services to Adults with Disabilities at the Dental Hospitals of the Medical University of Innsbruck, Austria"

_healthcare, 2024, doi:10.3390/healthcare12050503_

Round 1

Reviewer 1 Report

Comments and Suggestions for Authors

Dear authors

The manuscript entitled “Retrospective Evaluation of 20 Years of Outpatient Dental Services to Adults with Disabilities at the Dental Hospitals of the Medical University of Innsbruck, Austria” aimed to perform a retrospective descriptive study using the demographic data from patients with disabilities who received dental care at the Medical University of Innsbruck.

Although the paper brought a relevant topic to dental practice that is poorly discussed in the literature, improvements on the paper should be made so it could be considered for publication. Overall, the authors did not follow all the objectives proposed (patient and performance-related outcomes of their outpatient program) and should develop more discussion of their findings and how they could impact further studies. Other suggested modifications are included below.

INTRODUCTION

Lines 44-67 should be moved to the discussion section.

Is there any previous publication with a retrospective or prospective evaluation of dental services for disabled adults? If so, its main findings should be included.

(lines 71-73) Clarify what is these contributions: Is there any aspect missing from other studies discussed by Ferreira that were included in the present work?

MATERIAL AND METHODS

The distribution of the list of data items collected looks confused. Try to organize them in tables instead of bullet points.

Are there any exclusion criteria? Please clarify.

DISCUSSION

From the most common medical diagnosis in the study (intellectual disablement, epilepsies, seizures) is there any correlation with oral diseases that could explain the prevalence? Please discuss.

In the introduction, the authors suggested that one of the aims of the study is to evaluate the patient and performance-related outcomes of their outpatient program. How was this evaluation made? In many of the patients treated, no follow-up was made, as the authors mentioned.

The lack of follow-up raises the question of if the treatment provided was efficient, which is a major flaw in the study.

CONCLUSION

As the study is a descriptive retrospective study, the conclusion should include the main findings: what was the average profile of patients who received treatment? Age, gender, type of disabilities, treatment received, outcome, etc

Author Response

Dear Reviewer,

Reviewer 1’s comments.

Comments and Suggestions for Authors:

Dear authors

The manuscript entitled “Retrospective Evaluation of 20 Years of Outpatient Dental Services to Adults with Disabilities at the Dental Hospitals of the Medical University of Innsbruck, Austria” aimed to perform a retrospective descriptive study using the demographic data from patients with disabilities who received dental care at the Medical University of Innsbruck.

Although the paper brought a relevant topic to dental practice that is poorly discussed in the literature, improvements on the paper should be made so it could be considered for publication. Overall, the authors did not follow all the objectives proposed (patient and performance-related outcomes of their outpatient program) and should develop more discussion of their findings and how they could impact further studies. Other suggested modifications are included below.

Response: Dear reviewer

Thank you for the diligent review of our manuscript and for your constructive criticism.

Please find enclosed the changes that were made in response to each comment.

INTRODUCTION

Lines 44-67 should be moved to the discussion section.

Response: We moved lines 44-67 into the discussion.

Is there any previous publication with a retrospective or prospective evaluation of dental services for disabled adults? If so, its main findings should be included.

Response: Very few publications are available on chairside dental treatment of disabled individuals. We included two feasibility studies in the introduction and the discussion.  

(lines 71-73) Clarify what is these contributions: Is there any aspect missing from other studies discussed by Ferreira that were included in the present work?

Response: Ferreira et al. (2023) asserted that most publications on special healthcare needs are cross-sectional studies and simple literature studies, results of which are not directly applicable in clinical practice. Our contributions are clinical data (lines 61-72).

MATERIAL AND METHODS

The distribution of the list of data items collected looks confused. Try to organize them in tables instead of bullet points.

Response: We arranged the items more comprehensibly.

Are there any exclusion criteria? Please clarify.

Response: There were in- and exclusion criteria for participation in outpatient treatment. Please see lines 82- 92.

DISCUSSION

From the most common medical diagnosis in the study (intellectual disablement, epilepsies, seizures) is there any correlation with oral diseases that could explain the prevalence? Please discuss.

Response: A limited ability to cooperate, hampered access to dental care, and a complaint-driven rather than preventive utilization of dental services have been identified as reasons for poorer oral hygiene, poorer periodontal status conditions, and higher unmet caries treatment needs in individuals with intellectual disabilities. Please see lines 271-276.

In the introduction, the authors suggested that one of the aims of the study is to evaluate the patient and performance-related outcomes of their outpatient program. How was this evaluation made? In many of the patients treated, no follow-up was made, as the authors mentioned.

The lack of follow-up raises the question of if the treatment provided was efficient, which is a major flaw in the study.

Response: We reworded the purpose of our study in terms of aiming “a breakdown of chairside dental services to disabled adults by dental students over a period of 20 years in order to evaluate the performance-related outcomes of our outpatient program and to discuss feasibility of disabled individuals’ chairside treatment” (lines 61-64). Limitations of our study are addressed in lines 352-355.

CONCLUSION

As the study is a descriptive retrospective study, the conclusion should include the main findings: what was the average profile of patients who received treatment? Age, gender, type of disabilities, treatment received, outcome, etc

Response: We expanded the conclusion and included the main findings.

 We hope that you will find these changes satisfactory and deem the revised manuscript suitable for publication in healthcare.

 Thank you for your consideration.

With kind regards,

Reviewer 2 Report

Comments and Suggestions for Authors

Dear authors, I advise you to follow our advice to improve your manuscript:

I advise you to add more comprehensible graphics in the results

In the discussion you should add studies that have previously dealt with a similar topic.

The discussion is poor in studies that have dealt with a similar topic.

Limitations should be added in a separate chapter and not together with the discussion.

The bibliography should be improved by adding more bibliographic citations 

Additional comments:

- The scientific English of the article must be improved, the syntax must be improved and more appropriate English must be used.

- The graphs in the results need to be replaced and pie charts used so that they can be more easily understood.

- The discussion should be expanded, articles should be found in the literature that have discussed a similar topic in the past, and the articles should be compared with the manuscript.

- The conclusion should be expanded.

- The bibliography should be expanded and the most recent articles possible should be used.

Comments on the Quality of English Language

English needs to be checked and syntax improved.

Author Response

Dear Reviewer 2,

Reviewer 2’s comments.

 Comments and Suggestions for Authors:

Dear authors, I advise you to follow our advice to improve your manuscript:

I advise you to add more comprehensible graphics in the results

In the discussion you should add studies that have previously dealt with a similar topic.

The discussion is poor in studies that have dealt with a similar topic.

Limitations should be added in a separate chapter and not together with the discussion.

The bibliography should be improved by adding more bibliographic citations 

Response: Dear reviewer, 

Thank you for the diligent review of our manuscript and for your constructive criticism.

Please find enclosed the changes that were made in response to each comment.

Additional comments:

- The scientific English of the article must be improved, the syntax must be improved and more appropriate English must be used.

Response: We gave our best to improve the text with respect to appropriate English and syntax. However, time for revision was short. We thus focused on contents. If English is still considered unsatisfactory, we will employ scientific proofreading and editing support.

- The graphs in the results need to be replaced and pie charts used so that they can be more easily understood.

Response: We replaced the graphs with more comprehensible pie charts.

- The discussion should be expanded, articles should be found in the literature that have discussed a similar topic in the past, and the articles should be compared with the manuscript.

Response: Very few publications are available on chairside dental treatment of disabled individuals. We included two feasibility studies in the introduction and the discussion [References 11, 23, and follow-up study 24] and expanded the discussion. 

- The conclusion should be expanded.

Response: We expanded the conclusion.

- The bibliography should be expanded and the most recent articles possible should be used.

Response: We expanded the bibliography using the most recent articles available.

Limitations are addressed in a separate chapter.

We hope that you will find these changes satisfactory and deem the revised manuscript suitable for publication in healthcare.

Thank you for your consideration.

With kind regards,

Reviewer 3 Report

Comments and Suggestions for Authors

1. what is the basic objectives of the this study. After retrieving all this data, what do authors dissipate. What is the benefit of this study to the community? Why your study get published. What is the benefit? Is the study is benefit for the community? If yes, kindly elaborate

2. The data retrieved is of 20 years back. A lot of technical, infrastructure, professional changes/modifications/ improvements  have been occur in this era. do you think its wise to present such a data. 

3. Although write up is excellent but these are basic points which should be considered. The solution is that you can put these prespectives in the limitations and kindly explain in detail the objective of the study. 

4. Discussion presents their results. Which is ideally should not be the case. Kindly discuss results in discussion section in the light of other similar studies. 

Author Response

Reviewer 3’s comments.

Dear Reviewer,

Thank you for the diligent review of our manuscript and for your constructive criticism.

Please find enclosed the changes that were made in response to each comment.

Comments and Suggestions for Authors:

  1. what is the basic objectives of the this study. After retrieving all this data, what do authors dissipate. What is the benefit of this study to the community? Why your study get published. What is the benefit? Is the study is benefit for the community? If yes, kindly elaborate

Response: We reworded the purpose of our study in terms of aiming “a breakdown of chairside dental services to disabled adults by dental students over a period of 20 years in order to evaluate the performance-related outcomes of our outpatient program and to discuss feasibility of disabled individuals’ chairside treatment” (lines 61-64).

Ferreira et al. (2023) asserted that most publications on special healthcare needs are cross-sectional studies and simple literature studies, results of which are not directly applicable in clinical practice. Authors suggested that future research trends should focus on behavior guidance, dental education, and access to dental services. Our contributions are clinical data. Our study may serve as incentive for clinicians/researchers to report on treatment modalities and outcomes of chairside dentistry in patients with special oral health care needs, preferably by use of prospective study designs, to contribute data and strategies in the fight for control of oral health inadequacies. Please see lines 61-72.

  1. The data retrieved is of 20 years back. A lot of technical, infrastructure, professional changes/modifications/ improvements have been occur in this era. do you think its wise to present such a data. 

Response: We evaluated data from 2002 up to 2021. Our disabled persons’ outpatient program is still performed in the same way, as described in lines 240-261. Literature on chairside management of disabled persons is extremely scarce. Innovations are hardly reported.

  1. Although write up is excellent but these are basic points which should be considered. The solution is that you can put these prespectives in the limitations and kindly explain in detail the objective of the study. 

Response: Please see response to item 1 and our expanded conclusion.

  1. Discussion presents their results. Which is ideally should not be the case. Kindly discuss results in discussion section in the light of other similar studies.

Response: Very few publications are available on chairside dental treatment of disabled individuals. We included two feasibility studies in the introduction and the discussion [References 11 and 23, and follow-up study 24] and expanded the discussion.

For more details, please see the attached PDF file.

Response: In- and exclusion criteria for participation in outpatient treatment were specified. Please see lines 82- 92.

Medical diagnoses associated with disabilities are listed in Table 1.

Minor and major disabilities were categorized according to the Cooperation Level Scale [1].

We replaced the graphs with more comprehensible pie charts.

We hope that you will find these changes satisfactory and deem the revised manuscript suitable for publication in healthcare.

Thank you for your consideration.

With kind regards,

Round 2

Reviewer 1 Report

Comments and Suggestions for Authors

The authors provided the modifications requested

Reviewer 2 Report

Comments and Suggestions for Authors

Dear authors, you have followed my advice and increased the quality of your manuscript.

For me, there is no need for further changes

Congratulations

Reviewer 3 Report

Comments and Suggestions for Authors

Accepted